# Lysine Methyltransferase SMYD1 Regulates Myogenesis via skNAC Methylation

**DOI:** 10.3390/cells12131695

**Published:** 2023-06-22

**Authors:** Li Zhu, Mark A. Brown, Robert J. Sims, Gayatri R. Tiwari, Hui Nie, R. Dayne Mayfield, Haley O. Tucker

**Affiliations:** 1Department of Molecular Biosciences, The University of Texas at Austin, 1 University Station A5000, Austin, TX 78712, USA; lzhu3@stanford.edu (L.Z.); mark.brown@colostate.edu (M.A.B.); hnie2@mdanderson.org (H.N.); 2Department of Pathology, Lokey Stem Cell Research Building, 1291 Welch Rd Rm. G2035, Stanford, CA 94305, USA; 3Department of Clinical Sciences and Cell and Molecular Biology, Colorado State University, Fort Collins, CO 80523, USA; 4Flare Therapeutics, Cambridge, MA 02142, USA; 5Center for Biomedical Research Services, Department of Neuroscience, The University of Texas at Austin, 2500 Speedway A4800, Austin, TX 78712, USAdayne.mayfield@austin.utexas.edu (R.D.M.); 6Thoracic/Head and Neck Medical Oncology, MD Anderson Cancer Center, Houston, TX 77030, USA; 7Institute for Cellular and Molecular Biology, University of Texas at Austin, 1 University Station A5000, Austin, TX 78712, USA

**Keywords:** methyltransferase, transcriptional regulation, heart and skeletal muscle

## Abstract

The SMYD family is a unique class of lysine methyltransferases (KMTases) whose catalytic SET domain is split by a MYND domain. Among these, Smyd1 was identified as a heart- and skeletal muscle-specific KMTase and is essential for cardiogenesis and skeletal muscle development. SMYD1 has been characterized as a histone methyltransferase (HMTase). Here we demonstrated that SMYD1 methylates is the Skeletal muscle-specific splice variant of the Nascent polypeptide-Associated Complex (skNAC) transcription factor. SMYD1-mediated methylation of skNAC targets K1975 within the carboxy-terminus region of skNAC. Catalysis requires physical interaction of SMYD1 and skNAC via the conserved MYND domain of SMYD1 and the PXLXP motif of skNAC. Our data indicated that skNAC methylation is required for the direct transcriptional activation of *myoglobin* (*Mb*), a heart- and skeletal muscle-specific hemoprotein that facilitates oxygen transport. Our study revealed that the skNAC, as a methylation target of SMYD1, illuminates the molecular mechanism by which SMYD1 cooperates with skNAC to regulate transcriptional activation of genes crucial for muscle functions and implicates the MYND domain of the SMYD-family KMTases as an adaptor to target substrates for methylation.

## 1. Introduction

Myogenesis is a critical determinant of skeletal muscle development [1]. Key transcriptional regulators act as a complex and often compensatory network that regulate the spatial and temporal expression required for myogenic commitment, regeneration and homeostasis [1].

SET and MYND Domain 1 (SMYD1) are cardiac- and skeletal muscle-specific proteins [2]. Constitutive loss of *Smyd1* in mice results in embryonic lethality at E9.5, due to heart defects, including disrupted maturation of ventricular cardiomyocytes and malformation of the right ventricle [2]. Cardiac-specific and skeletal muscle-specific Smyd1 knockout mice further revealed the essential roles of Smyd1 in development and normal cardiac and skeletal muscle functions [3,4,5,6,7,8]. However, the molecular mechanisms underlying these observations remains largely unknown.

SMYD1 belongs to a unique class of KMTases [9] (Appendix A), whose catalytic SET domain is split by a MYND domain that contains a zinc finger motif involved in protein–protein interactions, suggesting that the SMYD family has unique substrate specificities. The roles of SMYD family proteins are dynamic, methylating a variety of histone and non-histone targets to regulate various cellular processes [9], making it difficult to reveal consistent patterns among the SMYD family members.

In contrast to other SMYD proteins, less is known about the SMYD1′s non-histone targets. SMYD1 has been shown to exclusively catalyze the tri-methylation of histone H3 lysine K4 (H3Km^3^) [10] with a single exception: the stress response factor Tribbles3/TRB3. When directly methylated by SMYD1, TRB3 acts as a co-repressor of SMYD1-mediated transcription during oxidative stress defense [5]. With its persistent expression throughout development and adulthood, SMYD1 plays a critical role in regulating myogenesis in both normal and pathological states [9]. Thus, it is informative to explore Smyd1 methylation targets during myogenesis.

*Smyd1* is encoded by three isoforms [9] (Figure 1A): two skeletal muscle and cardiac-specific isoforms, SMYD1a and SMYD1b; and the third, SMYD1c, which lacks the majority of the SET domain, is expressed exclusively in CD8+ cytotoxic T cells, where it regulates BCL2-meidated outer membrane integrity [11]. SMYD1a and SMYD1b differ only by a 13 amino acid insertion, making it difficult to distinguish between them biochemically. Studies in zebrafish have revealed the overlapping and unique roles of these isoforms in myogenesis [12,13,14,15]. Further studies are needed to analyze these isoforms and to determine their functional differences.

SMYD1 interacts with the skeletal muscle-specific splice variant of the α-subunit of the nascent polypeptide-associated complex (skNAC) [16]. skNAC is a transcription factor reported to enhance the transcription of myoglobin (MB), which encodes a heart- and skeletal muscle-specific hemoprotein that facilitates oxygen transport. Consistent with their physical interaction, the temporal and spatial expression patterns of skNAC were almost identical to those of SMYD1 [17]. *skNAC*-deficient mice die between E9.5 and E12.5 due, to cardiac malformations [17], including ventricular hypoplasia and decreased cardiomyocyte proliferation—phenotypes that are highly similar to those observed in *Smyd1* knockout mice [2,3,5]. Inhibition of skNAC or Smyd1 expression disrupts myofibrillogenesis and sarcomere formation [14,15,17]. Consistent with these observations, *Smyd1*- and *skNAC*-mutant embryos and C2C12 myoblasts exhibit similar gene dysregulation, including that of MB [5,14,15,17,18,19,20]. However, the molecular mechanisms underlying these similar phenotypes and gene expression patterns in *Symd1* and *skNAC* null were not clear.

To further reveal the relationship between SMYD1 and skNAC, we report here that both SMYD1a and SMYD1b methylate skNAC in vitro, whereas in vivo, skNAC methylation is performed exclusively by SMYD1b. SMYD1b methylates K1975 within the carboxyl region of skNAC, which requires their physical interaction via the conserved MYND domain of SMYD1 and the PXLXP motif of skNAC. Furthermore, our data suggest that methylation of skNAC is required for the activation of myoglobin. Our study provides a basic mechanistic insight into the cooperation of SMYD1 and skNAC in myogenesis and reveals the role of the MYND domain in substrate recognition and methylation. These results offer broad implications for identifying substrates and revealing catalytic patterns shared among SMYD family members.

## 2. Material and Methods

Cell culture and the production of stable cell lines. C2C12 and 293T cells were purchased from ATCC, and Dr. Gary Nolan supplied Phoenix A cells. Production of recombinant retroviruses and infection of cell lines were conducted with pBabePuro-SMYD1, pSilencer5.1-SMYD1 (shRNA), and pSilencer5.1-skNAC (shRNA). Briefly, 18–24 h prior to transfection, Phoenix A cells were plated at 1.5–2 million cells per 6 cm plate in growth media. The next day, the cells at 60–70% confluency were transfected with retroviral construct DNA using Fugene6 reagent. One day after transfection, the media was changed to a 3 mL fresh growth medium, and 80% confluent C2C12 was split at 1:8 or 80% confluent 10T1/2 cells at 1:12 per 6 cm plate. Supernatant from transfected Phoenix A cells was pipetted into 15 mL tubes and centrifuged at 1500 rpm for 5 min to pellet cell debris 48 h post-transfection. From each C2C12 or 10T1/2 plate, 1 mL of media was removed. A total of 3 μL polybrene (1000× at 5 mg/mL) was then added to each plate with gentle and thorough shaking, along with 1 mL of viral supernatant. Twenty-four hours post-infection, media on C2C12 or 10T1/2 plate was changed to a fresh growth medium. Virally infected cells were selected in a medium supplemented with 3 μg/mL puromycin 24–48 h post-infection. Cells were cultured for 5 days, with the selection medium replaced every 3 days. If the virus-infected cells reached 80% confluency, they were divided.

Bacterial and baculovirus expression constructs. pCI-skNAC was a generous gift from Dr. Rene St-Arnaud (Genetics Unit, Shriners Hospital, Montréal, QC, Canada). pCI-skNAC(BamHI) was a modified construct of pCI-skNAC with an insertion of BamHI and SalI restriction sites at the 5′ of skNAC via QuikChange Site-Directed Mutagenesis (Stratagene). pFast-Bac-SMYD1B was constructed by PCR using the 5′-primer (GCTCTAGAGCA CCATGGACGTGGAGGTCTTC) and the 3′-primer (CTCGAGCTGCTTCTTATGGAA CAG) with pBK-CMV-SMYD1B serving as a template. The PCR product was subcloned into pGEMT-easy, digested with Xba I/Xho I and then subcloned into pTP17 (a kind gift from the laboratory of Dr. Tanya Paull, University of Texas at Austin, Austin, Texas) after cutting with XhoI/Spe I. pFast-Bac-SMYD1B-(Y234F) was generated by QuikChange Site-Directed Mutagenesis (Stratagene) using pFast-Bac-SMYD1B as a template. pFast-Bac-HTb-skNAC was made by digesting pCI-skNAC with BamHI) to isolate the entire skNAC ORF. This fragment was then subcloned into pFast-Bac-Tb cut with BamHI. pGEX6P1-skNAC(1857-2187) was constructed by PCR with 5′-primer (5′-GGATCCCTTGTTAGCCCGCAAAAGGC-3′) and 3′-primer (5′-GCGGCCGCTTACATTG TTAATTCCAT-3′) using pCMV-Tag2b-SKNAC(1857-2187) as the template. The PCR product was subcloned into pGEMTeasy and digested with BamHI/Not I and then subcloned into pGEX-6P1 after cutting with BamH1I/Not I. pGEX6P1-skNAC(1857-2187) (L1952A) and pGEX6P1-SKNAC(1857-2187) (K-R) were generated by QuikChange Site-Directed Mutagenesis (Stratagene) using pGEX6P1-SKNAC(1857-2187) as templates. pGEX6P1-skNAC truncations were generated by PCR using pCI-skNAC as the template. The PCR products were digested with BamHI/Not I and subcloned into pGEX-6P1 after cutting with BamHI/Not I.

Retroviral small hairpin (sh)-RNA silencing constructs. pSilencer5.1-U6-SMYD1(shRNA), pSilencer5.1-U6-skNAC(shRNA) and pSilencer5.1-U6-Scramble(shRNA) were generated as follows: shRNA target sequences were selected with shRNA selector from Wistar institute. Ambion’s Insert Design Tool was used to convert the target sequences into hairpin shRNA-encoding DNA oligonucleotide sequences. These oligonucleotide sequences were annealed and ligated into pSilencer-5.1-Retro (Ambion).

Antibodies. Anti-SMYD1 3b2a monoclonal antibody (mAb) was described previously. Rabbit anti-skNAC polyclonal antibody (UT143) was generated with Cocalico Biologicals. Anti-FLAG M2 mAb is from Sigma (cat # F3165).

6xHis-tagged protein expression and purification. The Bac-to-Bac^®^ Baculovirus Expression Systems (Invitrogen) were used to construct 6xHis-tagged SMYD1. The full-length coding sequence for SMYD1 was cloned following XhoI/Spe I digestion into pTP17 (a kind gift from the laboratory of Dr. Tanya Paull, University of Texas, Austin, TX, USA), and the recombinant plasmid was transformed into DH10Bac™ competent cells (which contain the bacmid with a mini-attTn7 target site and the helper plasmid). Colonies containing recombinant bacmids were identified by disruption of the lacZa gene. High-molecular-weight mini-prep DNA was prepared from selected E. coli clones containing the recombinant bacmid, and this DNA was then used to transfect insect Sf21 cells to produce recombinant viruses. After two viral amplifications, 3–5 mL of viral supernatant was added to 100 mL of Sf21 cells at a concentration of 1.0 × 10^6^ cells/mL. The recombinant baculovirus-infected Sf21 cells were harvested 48 h post-infection by lysis in RIPA buffer (50 mM Tris at pH 8.0, 150 mM NaCl, 0.5% deoxycholate, 0.1% SDS, 1% NP-40, 1 mM PMSF, 5 μg/mL pepstatin, 10 μg/mL aprotonin, 5 μg/mL leupeptin, 1 mM benzamidine, 5 mM NaF and 5 mM NaOV4). Whole-cell extracts were clarified at 10,000 g for 10 min at 4 °C, and supernatants were incubated in batches with Ni^2^-NTA agarose (QIAGEN) by shaking (200 rpm on a rotary shaker) at 4 °C for 2 h. After incubation, the lysate-Ni-NTA mixture was loaded onto a column and washed twice with BC1000 buffer (20 mM Tris-HCl at pH 8.0, 1000 mM NaCl, 0.2 mM EDTA, 10% glycerol, 0.2 mM PMSF and 0.2% Tween 20), then washed with BC100 (20 mM Tris-HCl at pH 8.0, 100 mM NaCl, 0.2 mM EDTA, 10% glycerol, 0.2 mM PMSF and 0.2% Tween 20) twice. Bound proteins were eluted 6 times with 600 uL BC100 supplemented with 300 mM imidazole. The eluates were analyzed using SDS-PAGE. Eluates with high concentrations of proteins were combined and dialyzed against BC100. The dialyzed proteins were then employed in methyltransferase assays.

GST-fusion protein expression and purification.*E. coli* strain BL21 cells, which carry a recombinant pGEX6p1 plasmid, were grown in 2 mL LB medium (containing 100 ug/mL ampicillin) overnight. The next day, the entire overnight culture was added to 50 mL LB medium (containing 100 ug/mL ampicillin) and allowed to grow until the optical density at 600 nm reached 0.4 to 0.6. IPTG was then added to 0.2 mM, and the cultures were incubated for an additional 3 to 4 h. The cells were pelleted at 8000× *g* for 10 min at 4 °C. Bacteria were resuspended in 2 mL of PBS and 200 μg of lysozyme was added. After a 15 min incubation on ice, dithiothreitol to 5 mM and protease inhibitors to final concentrations of 0.1 mM PMSF, 100 μg of aprotinin per mL, 10 μg of leupeptin per mL, 10 μg of pepstatin per mL, 5 mM NaF, 5 mM NaVO4 and 1 mM benzamidine were added. Sarcosyl was added to a final concentration of 2%, and the bacterial suspension was sonicated for 30 s, left on ice for 1 min and then sonicated for an additional 30 s. Triton X-100 was then added to a final concentration of 4%, and the lysates were incubated with shaking at 4 °C for 20 min. The sample was centrifuged at 16,000× *g* for 10 min at 4 °C, then 0.5 mL 50% glutathione-Sepharose slurry (Pharmacia) was added to the supernatant. The mixture was then shaken at 4 °C for 2 h. The beads were centrifuged briefly and then washed with PBS 3 times. The bound proteins were eluted with elution buffer (50 mM Tris [pH 8.0], 100 mM NaCl, 10% glycerol, 1 mM DTT and 5 mM freshly added reduced glutathione). The elution employed a buffer volume equal to 1–2 times the packed bed volume at room temperature for 20 min. The beads were centrifuged briefly and the supernatant was collected. The elution was repeated, and the supernatants were combined.

In vitro lysine methyltransferase (KMTase) assays. In vitro methylation was carried out in a 40 μL reaction volume, including purified 6xHis-tagged-SMYD1 or immunoprecipitated SMYD1 bound to protein A-sepharose beads, 6xHis–tagged-skNAC or GST-skNAC truncations, and 1 μCi of S-adenosyl-[^3^H-methyl]-L-methionine (^3^H-AdoMet; 72 Ci/mmole; GE healthcare) were incubated at 30 °C for 1 h in a buffer containing 50 mM Tris-HCl (pH8.5), 5 mM MgCl2 and 4 mM DTT. The reactions were terminated by the addition of 4X SDS buffer. Histones were resolved by 12% SDS-PAGE and visualized using Coomassie blue R250 stain. In the 22% PPO solution, [^3^H]-methyl labeling was detected by fluorography. The dried gels were exposed to Kodak MRX films for various periods prior to autoradiography.

In vivo lysine methyltransferase (KMTase) assays. For in vivo methylation of skNAC(1857-2187), 293T cells were transiently co-transfected with pCMV-Tag2b- skNAC(1857-2187) and pBK-CMV-Smyd1 wt or SET domain mutants. At 48 h after the transfection, the cells were incubated with cycloheximide (100 mg/mL) and chloramphenicol (40 mg/mL) in normal DMEM growth medium (medium A) for 30 min. The medium was then replaced with 2.5 mL of medium B (Dulbecco’s modified Eagle’s medium without methionine, cysteine or glutamine [GIBCO, cat#21013], supplemented with penicillin, streptomycin, cysteine, glutamine and 10% fetal calf serum, which was dialyzed against modified DMEM [GIBCO, cat#21013]). L-[methyl3H]methionine was added to medium B at a concentration of 10 mCi per mL. Then, the cells were incubated for an additional 3 h and lysed in RIPA buffer (50 mM Tris at pH 34 8.0, 150 mM NaCl, 0.5% deoxycholate, 0.1% SDS, 1% NP-40, 2 mM NaF, 2 mM NaOV4 and protease inhibitor cocktail tablets [Roche Diagnostics] at two fold concentration). skNAC(1857-2187) was immunoprecipitated with an anti-FLAG antibody. The immunoprecipitated proteins were resolved using 8% SDS-PAGE. In a 22% PPO solution [3H] Methyl labeling was detected by autoradiography. The dried gels were exposed to Kodak MRX films for 1 week.

Gene expression analysis. Stably transduced or transiently transfected C2C12 cells were induced to differentiate for 48 h. Gene expression was then analyzed by RT-qPCR. Data represent the mean ± S.D. (*n* = 3). * *p*-value < 0.05, ** *p* < 0.01, *** *p* < 0.001 and ns = not significant (*p* > 0.05) (by *t*-test).

## 3. Results

SMYD1a and SMYD1b, but not SMYD1c, methylated skNAC in vitro. SMYD1 was observed to interact with skNAC, a transcriptional activator specific to the heart and skeletal muscles [16]. *skNAC* and *Smyd1* knockout mice shared similar defects in embryonic heart development and postnatal skeletal muscle growth [5,6,7,8,17]. These findings, combined with the notion that SMYD1 is a lysine methyltransferase, encouraged us to investigate whether SMYD1 could methylate skNAC.

FLAG-tagged SMYD1a, SMYD1b and SMYD1c (Figure 1A) were immunoprecipitated from transiently transfected 293T cells and used in in vitro KMTase assays using in vitro purified GST-skNAC (amino acid residues1857-2187; Figure 1B), a carboxyl-terminal fragment of skNAC previously shown to interact with SMYD1 in two hybrid assays [16] as a substrate. Both SMYD1a and SMYD1b robustly methylated skNAC(1857-2187) (Figure 1C). However, SMYD1c, which lacked a portion of the SET domain, did not methylate skNAC.

To confirm the methylation of skNAC by SMYD1, KMTase assays were performed using purified His-tagged SMYD1b and GST-tagged skNAC(1857-2187). GST-skNAC(1857-2187) was methylated by SMYD1b-6xHis (Figure 1D). Thus, the methylation of skNAC is a direct and intrinsic activity of SMYD1.

SMYD1b methylates skNAC in vivo. To address whether skNAC is methylated by SMYD1 in vivo. 293T cells were co-transfected with FLAG-skNAC(1857-2187) and SMYD1a or SMYD1b. Two days after transfection, cells were labeled with L-[methyl-^3^H]-methionine, then FLAG-skNAC(1857-2187) was immunoprecipitated, and [^3^H]-methyl labeling was detected by autoradiography. Surprisingly, compared to the in vitro methylation assay, in which FLAG-skNAC(1857-2187) was methylated by both isoforms, skNAC was methylated exclusively by SMYD1b in vivo (Figure 1E). The potential reason underlying the difference between the in vivo and in vitro results is addressed in the Discussion section.

We employed a pulse-chase approach to determine whether SMYD1 methylation may act to stabilize skNAC expression. We constructed a C2C12 cell line that stably expresses SMYD1 shRNA by retroviral transduction (detailed in the Materials and Methods section). Following a pulse with L-[methyl-3H]-methionine, we chased the newly synthesized skNAC over a time period of 8 h. As shown in Appendix A, no significant loss in skNAC protein expression was observed. These results suggested that SMYD1 may, instead, directly regulate skNAC-mediated transcription.

Both the SMYD1 catalytic domain and the interaction of SMYD1 and skNAC are required for skNAC methylation. SMYD1 contains a SET domain and a MYND domain. While SET domains are responsible for catalyzing lysine methylation [21], the MYND domain of SMYD1 binds to skNAC through the PXLXP motif of skNAC [16].

To determine whether the SET domain was required for skNAC methylation, the catalytically conserved Y234 was replaced with phenylalanine (indicated by the asterisk in Figure 2A). This SET domain mutant failed to methylate skNAC (Figure 2C). This result further indicated that SMYD1 has intrinsic methyltransferase activity, and that the SET domain of SMYD1 is required for this activity.

To determine whether the MYND domain of SMYD1 was required for skNAC methylation, the conserved cysteine residues required for zinc binding [22] were mutated to serine (Figure 2B). The mutations have been shown to abolish the ability of SMYD1 to bind skNAC [16]. The MYND domain mutant failed to methylate skNAC (Figure 2C). Therefore, the MYND domain of SMYD1 is required for the methylation of skNAC, likely by mediating the interaction between SMYD1 and skNAC.

To further confirm that the above interaction is indeed required for methylation, we employed a skNAC mutant in which the conserved leucine within PXLXP was substituted with alanine (Figure 2D). This mutant has been shown to abolish the ability of skNAC to bind to SMYD1 [16]. Compared with the methylation of wildtype skNAC, the methylation of the PXLXP mutant by SMYD1 was dramatically decreased (Figure 2E).

Together, these results indicate that the interaction between SMYD1 and skNAC via the conserved MYND domain of SMYD1 and the PXLXP motif of skNAC is essential for skNAC methylation. We have addressed the implication of this finding in revealing the catalytic pattern shared among SMYD family members in the Discussion section.

SMYD1b methylates K1975 of skNAC. We have demonstrated that a carboxyl-terminal fragment of skNAC(1857-2187), previously shown to interact with SMYD1 in two-hybrid assays [16], can be methylated by SMYD1 (Figure 1C–E). To narrow down the region(s) of skNAC modified by SMYD1, we performed methylation assays with the following three skNAC fragments (Figure 3A): skNAC(1857-1995), an N-terminal portion of skNAC(1857-2187) that contains the PXLXP motif but does not contain the *αNAC* region; skNAC(1996-2187), a C-terminal peptide that lacks the PXLXP motif but includes the *αNAC* region; and skNAC(1928-2000), a smaller N-terminal region of skNAC(1857-2187) that contains the PXLXP motif.

As shown in Figure 3B, SMYD1b methylated skNAC(1857-1995) and skNAC(1928-2000) but not skNAC(1996-2187). These results suggested that a methylation site exists within the carboxyl-terminal region (residues 1928-2000) of skNAC. Each of the 5 lysines contained within skNAC(1928-2000) (Appendix A) was substituted individually with arginine and tested in the methylation assay. As shown in Figure 3C, the mutation eliminated methylation at K1975, whereas the methylation of other K/R mutants was unaffected. Thus, SMYD1 methylates skNAC at K1975—a residue proximal to the PXLXP interaction motif.

SMYD1b, but not SMYD1a, regulated MB transcription in vivo. Myoglobin (MB), a muscle-specific hemoprotein, was one of the most down-regulated genes in previous analyses of both SMYD1 and skNAC mutant mice [5]. Thus, we analyzed the MB expression in C2C12 cells that stably expressed SMYD1 or skNAC shRNA. In these cells, SMYD1 or skNAC expression was efficiently silenced (Figure 4A,B). Compared to cells expressing a scrambled shRNA control, knockdown of SMYD1 or skNAC resulted in significantly reduced MB transcript levels (Figure 4C,D).

To gain further insight into the regulation of MB by SMYD1, we analyzed MB transcription in C2C12 cells that overexpress SMYD1a or SMYD1b. C2C12 cells were transiently transfected with either splice form and were then induced to differentiate. Two days later, MB was analyzed by RT-PCR. Compared to mock transfected (vector only) cells, MB mRNA levels were strongly elevated following overexpression of SMYD1b but not SMYD1a (Figure 4E). Note that the in vivo methylation assay (Figure 1E) also demonstrated that skNAC was methylated by SMYD1b but not SMYD1a. These results indicate that SMYD1 regulates MB transcription likely through skNAC methylation.

Next, we examined whether the SET and MYND domains of SMYD1b are required for the regulation of MB transcription. C2C12 cells were transiently transfected with SMYD1b WT as well as the SET and MYND domain mutants described above. As shown in Figure 4F, MB expression was upregulated only by SMYD1b WT but not by SMYD1 SET nor MYND domain mutants. These results indicate that the methyltransferase activity of SMYD1 is required for the regulation of MB transcription. They further indicate that the MYND domain-dependent interaction of SMYD1 and skNAC is necessary as well. Collectively, these results suggest that SMYD1 regulation of MB transcription is dependent on skNAC methylation.

## 4. Discussion

Sequence comparisons of the SMYD family members revealed that each of their SET domains lack a motif (YxG) that is strongly conserved among all other SET domain HMTases (Appendix A). This motif is essential for catalyzing methyl group transfer to a specific lysine on a histone tail. This observation, combined with another unique feature of this family—that the SET domain is split by a MYND domain [9]—suggested that SMYD family proteins have unique substrate specificities.

Indeed, we and others [23,24,25,26] have identified a number of non-histone proteins whose methylation by SMYD2 has led to dramatic alterations in their activity. In an attempt to search for non-histone substrates of SMYD1, we tested the hypothesis that SMYD1 methylates the proteins with which it interacts. Such a protein is skNAC [5,17,27], a heart and skeletal muscle-specific transcriptional activator of myoglobin [5,27]. The present study revealed the critical nature of SMYD1 isotype specificity, the identification of SMYD1-mediated lysine methylation within the carboxyl region of skNAC, the requirement for their interaction in this process, and the transcriptional regulation of myoglobin by SMYD1 likely through skNAC methylation.

SMYD1 muscle isoforms and myoglobin regulation. Alternative pre-mRNA splicing allows individual genes to produce multiple protein isoforms, thereby serving as major contributors to protein diversity in higher eukaryotic organisms. SMYD1 is expressed in two skeletal muscle- and cardiac-specific isoforms. SMYD1a and SMYD1b differed by a 13 amino acid insertion within the SET domain due to the alternative inclusion of exon 5 (Figure 1A). Although both SMYD1a and SMYD1b methylated skNAC in vitro (Figure 1C), only SMYD1b could methylate skNAC in vivo (Figure 1E). Consistent with this, only SMYD1b overexpression activated MB transcription (Figure 4E).

The orthologous isoforms in zebrafish, Smyd1a and Smyd1b, exhibit distinct patterns of expression [10]. Smyd1a transcripts were first detected at 6 h post-fertilization, and their abundance increased significantly during somitogenesis. In contrast, Smyd1b was expressed 5 h later than SMYD1a, suggesting that their alternative splicing is regulated during development. Furthermore, the zebrafish study [14,15] reported that the KD of Smyd1b resulted in cardiac and skeletal muscle defects due to disturbed myofibril assembly, whereas SMYD1a depletion had no effect. Our observation that SMYD1b, but not Smyd1a, methylates skNAC in vivo provides a potential molecular explanation for the different roles of SMYD1a and SMYD1b in myofibril organization.

As shown in Appendix A, the sequence constituting the 13 residue insertion of exon 5 is relatively conserved among SMYD1b vertebrate orthologs—particularly at potential phosphorylation sites (e.g., Ser217, Thr221 and Ser225). Mutation of each of these sites had little to no effect on sarcomere association [12]. However, it remains to be determined whether these sites are phosphorylated in vivo or whether other conserved regions of the 13mer have functional consequences. Particularly curious is the conserved tetramer (FHSQ) at the C-terminal end of exon 3. FHSQ is identical to a conserved immunodominant motif within the V3 region of HIV gp125 [28]. Molecular docking analyses of both HIV gp125 [29] and SMYD1 [30] indicated that, in both molecules, FHSQ is accessible and completely solvent exposed.

These results indicated that SMYD1a and SMYD1b have distinct functions in myogenesis. It is possible that the methyltransferase activity of SMYD1a is temporally inhibited by some factors in vivo through a 13 amino acid insertion in the SET domain. These factors were not present in in vitro HMTase assays with purified and immunoprecipitated proteins. It will be informative to determine how the 13 amino acid insertion within the SET domain of SMYD1a promotes enzymatic differences, potentially via selective recruitment of protein effectors.

The MYND domain of SMYD1 is essential for skNAC methylation. Our results demonstrate that the MYND domain of SMYD1 and the PXLXP motif in skNAC are essential for the methylation of skNAC by SMYD1 (Figure 2C,E). Thus, direct interaction between SMYD1 and skNAC is necessary for catalysis to occur. The structural basis for the interaction between skNAC and SMYD1, provided by the high-resolution SMYD1 crystal structure [30], is consistent with the biochemistry determined here. Modeling indicated that the SMYD1 MYND domain is the primary protein interface that partners with skNAC to regulate cardiomyocyte growth and maturation.

The MYND protein–protein interaction domain is primarily comprised of a C4C2HC-type zinc finger [31]. It was previously identified within transcriptional regulators, such as AML1/ETO, and in the tumor suppressor BS69 [22]. The MYND domain of SYMD1 has been shown to interact with the PXLXP motif of skNAC [16], i.e., akin to an adaptor that brings SMYD1 and skNAC together. Structural modeling [30] was also performed on the MYND domain of AML1/ETO in complex with a PXLXP-containing peptide. While sharing <30% sequence identity, both SMYD1 and PXLXP peptides were virtually superimposed within a shallow, fully solvent-exposed surface groove readily accessible to skNAC [30].

The *Smyd* family, a special class of KMTases whose catalytic SET domain is split by a MYND domain, consists of five members (Smyd1-5) [9,32] (Appendix A). Emerging evidence has revealed the diverse and critical roles of these family proteins in development and tumorigenesis [9,32]. In addition to SMYD1, SMYD2 has been shown to bind multiple proteins via the PXLXP motif [33].

These studies suggest a model in which the MYND domain of SMYD MTases recruits substrates containing PXLXP to prepare them for methylation through orientation and/or structural transformation. This model has broad implications, including the prediction/identification of potential novel substrates, revealing the catalytic patterns shared among *Smyd* family members and decoding the underlying molecular mechanisms that govern *Smyd* functions.

Activation of myoglobin transcription by SMYD1 and skNAC. skNAC was shown to activate luciferase expression driven by the *Mb* promoter [27]. Smyd1 and skNAC mutant embryos exhibited similar cardiac malformation phenotypes and gene dysregulation, including that of MB (6). However, the molecular mechanisms underlying these similar phenotypes and gene expression patterns in SYMD1 and skNAC null mice were not clear.

SMYD1b, but not Smyd1a, methylated skNAC in vivo (Figure 1E). Accordingly, overexpression of SMYD1b, but not SMYD1a, activated MB expression (Figure 4E). Further analyses showed that MB mRNA levels were upregulated only by SMYD1b WT but not by SET nor MYND domain mutants (Figure 4F). Together, these results strongly suggest that SMYD1 regulates MB expression through skNAC methylation.

MB is an abundant cytoplasmic hemoprotein that is expressed in cardiomyocytes and oxidative skeletal myofibers of vertebrates [34,35]. MB serves to store oxygen and to facilitate its transfer from the capillaries to the mitochondria. MB scavenges reactive oxygen species, thereby serving as a critical cytoprotective protein that limits the toxic effects of oxidative stress in striated muscles [36]. Targeted deletion of *Mb* in mice [37] results in partial embryonic lethality at ~E10.5, with cardiac defects highly similar to those observed in *skNAC* nulls [17] and *Smyd1* heart-conditional [3,5] KO mice. Our study provides a molecular explanation for at least some of the phenotypes shared by *Smyd1*, *skNAC* and *Mb* knockout embryos.

Searching mouse promoter databases revealed numerous genes containing skNAC binding sites. As we demonstrated directly in a recent publication [19], global gene expression analyses of differentiating C2C12 myoblasts following shRNA-mediated knockdown of SMYD1 or skNAC not only confirmed MB as a high-priority target but also identified a large array of additional SMYD1 and skNAC-regulated genes. Future studies will evaluate the effect of skNAC methylation by Smyd1 on the expression of these target genes.

## 5. Conclusions

In this study, we identified the skNAC transcription factor as a methylation target of SMYD1. We established the site of methylation and its effects on skNAC-mediated transcription. Our study revealed the molecular mechanisms underlying the cooperation between SMYD1 and skNAC in myogenesis.

## Figures and Tables

**Figure 1 cells-12-01695-f001:**
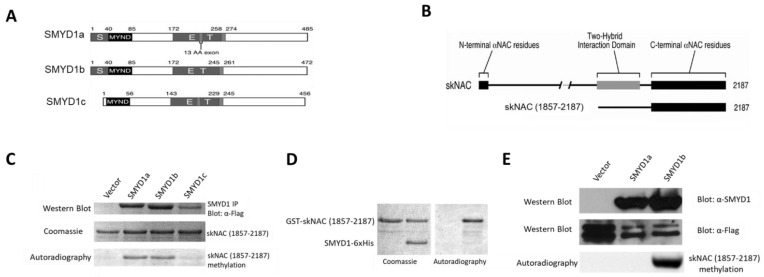
**Both SMYD1a and SMYD1b methylate skNAC in vitro, but only SMYD1b methylates skNAC in vivo.** (**A**) Schematic representation of SMYD1 orthologs. The SET domain was split into the S and ET portions by the MYND domain. SMYD1a has a unique exon 6 that encodes a 13 amino acid insertion, which shares no sequence identity with known proteins. SMYD1c transcription is initiated ~160 bp upstream of exon 1 of the other isoforms and is alternatively spliced so as to eliminate the N-SET domain. The positions of the amino acids are indicated. (**B**) Schematic of full-length skNAC and its sub-region, skNAC(1857-2187), originally identified in a SMYD1 yeast two-hybrid screen [11]. Sequences common to α-NAC are denoted by black boxes (23 amino-terminal and 192 carboxyl-terminal residues). The two-hybrid interaction domain is represented by a grey box. (**C**) Both SMYD1a and SMYD1b methylate skNAC in vitro. FLAG-SMYD1a, FLAG-SMYD1b or FLAG-SMYD1c were immunoprecipitated from transiently transfected 293 T cells and used in in vitro KMTase assays, as described in the Materials and Methods section. Two micrograms of purified GST-skNAC(1857-2187) were used as the substrate. The reaction products were separated by SDS-PAGE, stained with Coomassie and subjected to autoradiography. Western blot analysis (top panel) was performed with an anti-FLAG antibody using 10% of the immunoprecipitated SMYD as the input. Equal loading of GST-skNAC (middle panel) was used for the KMTase assay. Methylation was identified by the autoradiography of incorporated 3H-S-Adenyl-Methione (SAM) (bottom panel). (**D**) SMYD1 directly methylated skNAC in vitro. KMTase assays were performed with purified His-tagged SMYD1b as enzyme. As substrates, 2 ug of GST-skNAC were used. The reaction products were separated using SDS-PAGE, stained with Coomassie (left) and subjected to autoradiography (right). (**E**) SMYD1b, but not SMYD1a, methylates skNAC in vivo. 293T cells were co-transfected with FLAG-tagged skNAC(1857-2187) and SMYD1a or SMYD1b. Two days after transfection, cells were labeled with L-[methyl-3H]-methionine in the presence of protein synthesis inhibitors. skNAC(1857-2187) was then immunoprecipitated with anti-FLAG, resolved by SDS-PAGE and subjected to autoradiography. Top panel: Western blot analysis was performed with anti-SMYD1 using the whole-cell extract as the input. Middle panel: Western blot analysis was performed with an anti-FLAG antibody using 10% of immunoprecipitated FLAG-tagged skNAC(1857-2187) as input. Bottom panel: Methylation was identified by autoradiography of incorporated 3H-S-Adenyl-Methione (SAM).

**Figure 2 cells-12-01695-f002:**
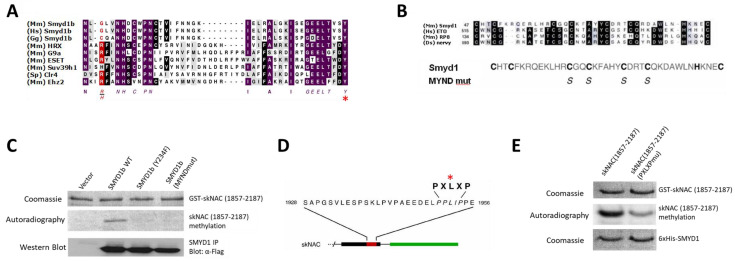
**The SET domain of SMYD1 and the interaction of SMYD1 and skNAC are required for skNAC methylation.** (**A**) Comparison of the C-terminal regions of the SET domains present in SMYD1 and other HMTases. The highly conserved NHxCxPN and GEELxxxY motifs of SET domains, which are strictly essential for KMTase activity, are shown in purple. In the SET domain mutant of SMYD1, the highly conserved tyrosine (Y234) (*), which is essential for SET MTase activity, was mutated. (**B**) Comparison of the MYND domain of mouse SMYD1 with other MYND domain-containing proteins (upper panel) was performed. In the MYND domain mutant of SMYD1, the highly conserved Cys residues, that are required for zinc binding, were mutated to S (lower panel). (**C**) The SET and MYND domains of SMYD1 are required for skNAC methylation. FLAG-tagged SMYD1b WT, SMYD1b-MYNDmut or SMYD1b-SETmut(Y234F) were immunoprecipitated with anti-FLAG Ab from transiently transfected 293T cells and used for in vitro KMTase assays. Purified GST-skNAC(1857-2187) was used as a substrate. Equal amounts of skNAC (top panel) were used in the KMTase assay. The middle panel shows an autoradiograph of GST-skNAC from the methylation reaction. Western blot analysis (bottom panel) was performed with anti-FLAG Ab using 10% of the immunoprecipitated SMYD1b as the input. (**D**) skNAC PXLXP mutant (L1952A) in which the L of PXLXP motif was replaced by A. This mutant was previously shown to abolish skNAC binding to SMYD1 in immunoprecipitation assays [16]. (**E**) KMTase assay was performed with purified his-tagged SMYD1b as enzyme. As substrates, 2 ug of GST-skNAC(1857-2187) WT or GST-skNAC(1857-2187) (L1952A) were used.

**Figure 3 cells-12-01695-f003:**
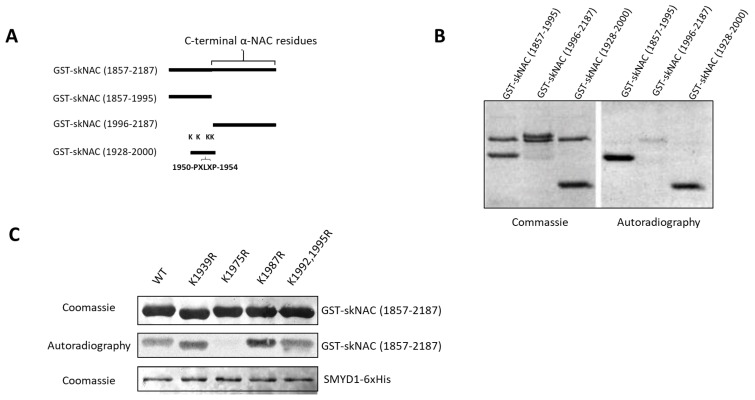
**Identification of the lysine residue of skNAC methylated by SMYD1.** (**A**) skNAC truncations employed in this study. Two truncations include an amino-terminal fragment of skNAC(1857-2187) that contains the region of skNAC encompassing residues 1857-1995, which are unique to skNAC, and a C-terminal fragment in the region encompassing skNAC residues 1996-2187, which are shared with α-NAC. Lysine (K) residues tested as putative targets of SMYD1-mediated methylation are noted on GST-skNAC (1928-2000). (**B**) KMTase assays were performed with purified His-tagged SMYD1b as enzyme. As substrates, 2 ug of GST-skNAC fragments were used. (**C**) K1975 of skNAC is methylated by SMYD1. Lysines (K) within skNAC(1928-2000) were mutated individually. GST-skNAC(1928-2000) containing these mutations were used as substrates in KMTase assays with purified His-tagged SMYD1b as enzyme.

**Figure 4 cells-12-01695-f004:**
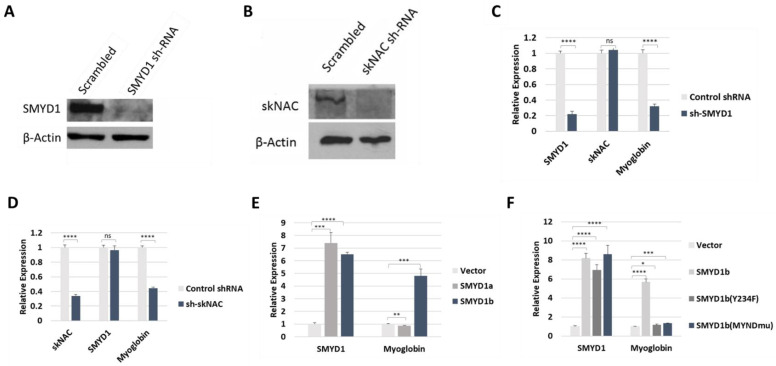
**skNAC and SMYD1 regulate myoglobin transcription.** (**A**) SMYD1 expression was efficiently knocked down by shRNA. C2C12 cells that stably expressed scrambled shRNA or SMYD1 shRNA were induced to differentiate for 48 h. The cells were then lysed, and whole-cell extracts were analyzed by Western blotting using anti-SMYD1 Ab. (**B**) skNAC expression was efficiently knocked down using shRNA. C2C12 cells stably expressing scrambled shRNA or skNAC were induced to differentiate for 48 h. The cells were then lysed, and whole-cell extracts were analyzed by Western blot with anti-skNAC Ab. (**C**) Myoglobin mRNA was significantly reduced in cells stably expressing SMYD1 shRNA. C2C12 cells stably expressing scrambled shRNA or SMYD1 shRNA were induced to differentiate for 48 h. SYMD1 and myoglobin expression in these cells were analyzed using RT-qPCR. (**D**) Myoglobin expression was significantly down-regulated in C2C12 cells stably expressing skNAC shRNA but not in scrambled shRNA controls. Following 48 h of induction, skNAC and myoglobin expression were analyzed by RT-qPCR. (**E**) SMYD1b, but not SMYD1a, regulated MB expression in vivo. C2C12 cells were transiently transfected with SMYD1a, SMYD1b or empty vector. At 24 h after transfection, the cells were induced to differentiate for an additional 48 h, and myoglobin expression was analyzed using RT-qPCR. (**F**) SET and MYND domains of SMYD1 are required for the regulation of myoglobin expression. C2C12 cells were transiently transfected with SMYD1b WT, SET domain mutant, MYND domain mutant or empty vector. At 24 h after transfection, the cells were induced to differentiate for 48 h. Myoglobin expression was then analyzed by RT-qPCR. Data represent mean ± S.D. (n = 3). * *p*-value < 0.05, ** *p* < 0.01, *** *p* < 0.001, **** *p* < 0.0001, ns = not significant (*p* > 0.05) (by *t*-test).

## Data Availability

Data is contained within the article or Appendix A.

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
