# Peer review of "Lysine Methyltransferase SMYD1 Regulates Myogenesis via skNAC Methylation"

_cells, 2023, doi:10.3390/cells12131695_

Round 1

Reviewer 1 Report

Zhu et al. analyzed crosstalk between the skNAC and Smyd1 transcription factors. Their central finding is that skNAC is a Smyd1 methylation target and that this methylation is important for skNAC’s function as a transcription factor.

While the story itself is interesting, there are several flaws that should be addressed:

1. The paper has not been written very carefully and should be thoroughly worked over.

a few examples, just from title and abstract:

- methyl transferase / methyltransferase

- abstract l.3: KMTases instead of KMTase

- l.4/5 – What does this sentence mean?

- several blanks after “exception“

- “carboxyl region“ – carboxy-terminus?

- the PXLXP motif (but some “the“s are also too much)

Layout of panels in figures is very inconsistent, some of them look as if copied from a poster.

Lots of typos: “SMYND1“ etc.

2. Fig.1

C:

Why do FLAG-tagged Smyd1 proteins run at a smaller MW?

Legend says that blot was anti-FLAG, however, the top panel is labelled aSmyd.

Why is there no Western for skNAC?

fluorography (image) or autoradiography (legend)?

D:

Coomassie

3. Fig.3

C (legend): 1975

4. Fig.4

Does SMYD1 knockdown affect skNAC levels or vice versa?

5. Further questions:

- The isoform issue is not really clear to me:

If Smyd1a does not methylate skNAC in vivo – why should it do so in vitro? I can understand that methylation might not take place in an organism (such as zebrafish), if, as the authors state, the timing is such that Smyd1b comes up later (and maybe also skNAC? – so that Smyd1a and skNAC might never “meet“). However, this should be completely irrelevant for their tissue culture model system.

If, as the authors state, Smyd1b comes up later during development – does this timing correlate with MB expression?

- What about the role of skNAC and Smyd1 in other areas of myogenesis, such as sarcomerogenesis – is it likely that skNAC methylation by Smyd1 is important for all of these, too (i.e. not just in the context of MB expression)?

- What about other transcriptional targets of skNAC-Smyd1 – does their expression also depend on skNAC methylation by Smyd1?

- Myoglobin is expressed in slow fibers only. There has been a conflicting debate with regard to the role of Smyd1-skNAC in the context of fiber type specification in the last 10-15 years. The authors should refer to these conflicting results and evaluate them in the context of their data.  

needs to be thoroughly edited

Author Response

Comments and Suggestions for Authors

Zhu et al. analyzed crosstalk between the skNAC and Smyd1 transcription factors. Their central finding is that skNAC is a Smyd1 methylation target and that this methylation is important for skNAC’s function as a transcription factor.

While the story itself is interesting, there are several flaws that should be addressed:

  1. The paper has not been written very carefully and should be thoroughly worked over.

a few examples, just from title and abstract:

- methyl transferase / methyltransferase

- abstract l.3: KMTases instead of KMTase

- l.4/5 – What does this sentence mean?

- several blanks after “exception“

- “carboxyl region“ – carboxy-terminus?

- the PXLXP motif (but some “the“s are also too much)

As suggested, we went through the manuscript carefully. We have revised all errors in our writing and our figures, including all the errors Reviewer 1 had pointed in his/her first comment.

Layout of panels in figures is very inconsistent, some of them look as if copied from a poster.

The panel layouts of all figures have been modified to make them more consistent.

Lots of typos: “SMYND1“ etc.

This as well as all other “inconsistences” have been corrected in the revised manuscript. This has been accomplished by going over the manuscript carefully, including revision of typos and other writing errors in texts or in figures.

  1. Fig.1C:

Why do FLAG-tagged Smyd1 proteins run at a smaller MW?

We apologize to the reviewer for what appears to be a molecular size contradiction. We suspect the impression resulted from an error in our Figure. In the now revised Figure 1C, we have labelled the top panel of Figure 1C as “Blot: α-Flag”.

As stated in the legend of Figure 1C, FLAG-SMYD1a, FLAG-SMYD1b, or FLAG-SMYD1c were immunopreciptiated from transiently transfected 293T cells by Flag antibody and used in in vitro KMTase assays. Western blot analyses (top panel) was performed with anti-FLAG antibody using 10% of the immunoprecipitated SMYD as input.

We apologize for the confusion, but the slower migrating bands in the Western blot analysis (which are also present in the empty vector control) are non-specific bands produced by blotting with anti-Flag antibody. Those bands should not have been included in the original Figure 1. Thus, our revised Fig. 1C contains only SMYD1-specific bands.

Legend says that blot was anti-FLAG, however, the top panel is labelled aSmyd.

We agree with the Reviewer and the top panel is now labelled “Blot: α-Flag” as appropriate.

Why is there no Western for skNAC?

We disagree with the Reviewer that Westerns for skNAC need to be shown.  2 ug of purified GST-skNAC(1857-2187) was used as substrate. The purified GST-skNAC is revealed following SDS-PAGE and staining with Coomassie (middle panel of figure 1C). Thus, it was unnecessary to perform an anti-skNAC Western blot.

fluorography (image) or autoradiography (legend)?

We used Fluorography to represent the lysine methylation reaction in which [3H]-methyl is used by methyltransferase to methylate substrate proteins.  

Autoradiography is the procedure to visualize radiolabeled proteins by fluorography.  During the procedure, gels were stained with Coomassie blue, destained and dried and exposed to Kodak film.

To clarify for the Reviewer and to be more consistent throughout the revised text, the label “Fluorography” is replaced by “Autoradiography” in all Figures.

D:Coomassie

The spelling has been corrected.

  1. Fig.3C (legend): 1975

We appreciate the Reviewer’s close attention, and this error has been corrected.

  1. Fig.4

Does SMYD1 knockdown affect skNAC levels or vice versa?

The Reviewer poses a good question. In response, we checked skNAC levels following SMYD1 knockdown. SMYD1 knockdown had no significant effect skNAC level (Fig.4C).

We also checked SMYD1 levels in skNAC knockdown. skNAC knockdown had no significant effect SMYD1 levels (Fig.4D).

  1. Further questions:

 - The isoform issue is not really clear to me:

If Smyd1a does not methylate skNAC in vivo – why should it do so in vitro? I can understand that methylation might not take place in an organism (such as zebrafish), if, as the authors state, the timing is such that Smyd1b comes up later (and maybe also skNAC? – so that Smyd1a and skNAC might never “meet“). However, this should be completely irrelevant for their tissue culture model system.

If, as the authors state, Smyd1b comes up later during development – does this timing correlate with MB expression?

Reviewer 1 raises an interesting point. We also found this interesting and as the Reviewer reiterates, we speculated about this issue in the Discussion section. To reiterate:

Smyd1 expresses two skeletal muscle and cardiac-specific isoforms which differ by a 13 amino acid insertion within the SET domain of SMYD1b due to alternative inclusion of a small exon by pre-mRNA splicing.

As noted by the Reviewer, both SMYD1a and SMYD1b methylate skNAC in vitro, whereas in vivo, skNAC methylation is performed exclusively by SMYD1b. One speculation that we provide is that the methyltransferase activity of SMYD1a is temporally inhibited by factors present in in vivo (but not in vitro) through the 13 amino acid insertion in SET domain. Obviously such a hypothetic factor(s) might not be present in the in vitro HMTase assays employed using purified and immunopreciptiated proteins.

Consistent with our observations, myosin heavy-chain protein accumulation and sarcomere organization are dramatically reduced in smyd1b mutant, but not in smyd1a mutant [14,15]. Similar defects also were observed in cardiac muscles of the smyd1b mutant, indicating that Smyd1b, but not Smyd1a, plays a key role in myosin heavy-chain protein expression and sarcomere organization in craniofacial and cardiac muscles [14,15]. Our observation that SMYD1b, but not SMYD1a methylated skNAC in vivo provides a potential molecular explanation for the different role of SMYD1a and SMYD1b in sarcomere organization.

We have revised discussion section to further clarify Reviewer 1’s question.

Please note that [xxx] are the corresponding papers referenced in our manuscript.

- What about the role of skNAC and Smyd1 in other areas of myogenesis, such as sarcomerogenesis – is it likely that skNAC methylation by Smyd1 is important for all of these, too (i.e. not just in the context of MB expression)?

Reviewer 1 makes another very good point. It was previously shown and stated in our Introduction and Discussion section that Inhibition of either skNAC or Smyd1 expression disrupted myofibrillogenesis and sarcomere formation [14,15,17]. skNAC-deficient mice die between E9.5 and E12.5 due to cardiac malformations [16], that are highly similar to those observed in Smyd1 knockout mice [2, 3, 5]. Consistent with these observations, Smyd1 and skNAC mutant embryos and C2C12 myoblasts exhibit similar gene dysregulation, including that of MB [5, 14, 15, 17-20]. We argue that our study provides a molecular rationale for these similar phenotypes and gene expression pattern.

Re the interesting question posed by Reviewer 1 of target genes other than MB, skNAC is a transcription factor that has been shown to bind to a specific DNA motif [24]. Searching mouse promoter databases, we found numerous genes containing skNAC binding site. As we showed directly in a recent publication [19], global gene expression analyses of differentiating C2C12 myoblasts following shRNA-mediated knockdown of SMYD1 or skNAC not only confirmed MB as a high priority target, but also identified a large array of additional SMYD1 and skNAC regulated genes.

These observations along with our own studies indicate that SMYD1 acts upstream to and via methylation of skNAC to regulate subsets of muscle-specific genes essential for myogenesis.

- What about other transcriptional targets of skNAC-Smyd1 – does their expression also depend on skNAC methylation by Smyd1?

Again another interesting point has been raised by Reviewer 1. While we have confirmed targets coregulated by SMYD1 and skNAC in several contexts, we do not know whether SMYD1-mediated methylation of skNAC is required in these cases. We have added this interesting point in the context of future studies in the revised Discussion section.

- Myoglobin is expressed in slow fibers only. There has been a conflicting debate with regard to the role of Smyd1-skNAC in the context of fiber type specification in the last 10-15 years. The authors should refer to these conflicting results and evaluate them in the context of their data.  

As suggested by Reviewer 1, we have referenced the papers studying the role of Smyd1-skNAC in the context of fiber type specification in the revised Introduction and Discussion.

Knockout or inhibition of skNAC expression in oxidative soleus muscle, resulted in reduced expression of oxidative fiber markers and enhanced expression of glycolytic fiber markers [16, 20]. While oxidative fiber markers were upregulated, glycolytic fiber markers were reduced in skNAC siRNA-transfected C2C12 cells. These seemingly conflicting observations indicate that the action of skNAC on skeletal muscle metabolic properties is dependent on cellular background.

Since we employed only C2C12 myoblasts for the present analysis, to evaluate the role of skNAC methylation by SMYD1 in fiber type specification was beyond our scope.

Reviewer 2 Report

In this manuscript, Zhu and colleagues show that skNAC is a target of SMYD1 methylation, illustrating its molecular mechanism.

The work is interesting.

The authors should add information about the number of experiments performed, the statistics applied to the analyses performed with p-value.

Figure 1C is not clear, SMYD1 IP and a-SMYD1 blot have different weights. Could the authors explain the weight difference on the same gel?

Author Response

In this manuscript, Zhu and colleagues show that skNAC is a target of SMYD1 methylation, illustrating its molecular mechanism.

The work is interesting.

We thank Reviewer 2 for this comment.

The authors should add information about the number of experiments performed, the statistics applied to the analyses performed with p-value.

As suggested by Reviewer 2, we added information about the number of experiments performed, the statistics applied to the analyses performed with p-values both in Materials and Methods and in Figure Legends.

Figure 1C is not clear, SMYD1 IP and a-SMYD1 blot have different weights. Could the authors explain the weight difference on the same gel?

We thank Reviewer 2, as we agree a clarification is in order.  Reviewer 1 also had issues with the description of these data (addressed above in response to Reviewer 1).

In Figure 1C, FLAG-SMYD1a, FLAG-SMYD1b, or FLAG-SMYD1c were immunopreciptiated from transiently transfected 293T cells by Flag antibody and used in in vitro KMTase assays. Western blot analyses (top panel) was performed with anti-FLAG antibody using 10% of the immunoprecipitated SMYD as input.

The top panel of Figure 1C should be labelled Flag antibody. It has been corrected.

The slower migrating bands in the Western blot analysis of the top panel (which were also present in the empty vector control) are non-specific bands from blotting with Flag antibody. We have eliminated them from the revised Figure1C, so that the blot only shows the SMYD1 specific bands.

Round 2

Reviewer 1 Report

All my questions have been sufficiently answered.

Some minor editing required (can be performed in the context of proofreading).